# DRMLP: Dynamic Regularized Multi-Layer Perceptron for Neural Granger Causality Discovery with Adaptive Temporal Penalties

## Abstract

With the rapid development of IoT devices, collecting multivariate time series data has become increasingly convenient. Understanding the causal relationships among different time series variables is critical for validating causal discovery methods and benchmarking their ability to recover ground-truth interactions in controlled synthetic environments. However, existing Granger causality approaches based on neural networks typically require modeling each time series variable separately and assume that the influence of historical values decays over time. These limitations result in complex models and poor performance in discovering causality in time series with long-range dependencies. To address these drawbacks, this paper proposes a model called DRMLP: Dynamic Regularized Multi-Layer Perceptron, a Granger causality discovery method capturing periodic temporal dependencies from the input weights of a convolutional network. The proposed approach employs a dual-branch neural network architecture: a linear causal discovery network is utilized to extract causal relations from sampled weight data, while a hierarchical regularization strategy is introduced to optimize the weights of the convolutional network. This design enhances the accuracy of causal relation discovery and reduces noise interference, thereby ensuring the temporal consistency of the identified causal structures. Experiments conducted on simulated datasets and real-world system-generated datasets show that DRMLP outperforms state-of-the-art baseline methods.

## 1 Introduction

The exploration of causal relationships among variables in multidimensional time series data is crucial to accurately predict outcomes and conduct intervention analyzes. For example, in neuroscience, brain activity propagates across different regions (Vicente et al., 2011), leading to fluctuations in various indicators as the activity spreads. Understanding the internal structure of the data and their propagation dynamics is particularly critical for predicting brain activity and informing therapeutic interventions. During the past few decades, researchers have made significant strides in uncovering causal relationships (Runge, 2018) from observational time series data (Gerhardus and Runge, 2020; Tank et al., 2021; Khanna and Tan, 2019; Pamfil et al., 2020) Granger causality analysis (Granger, 1969; Marinazzo et al., 2008) quantifies whether past values of one series help predict the future values of another.

Granger causality is widely used for time-series analysis and can be estimated via model-based or non-model-based methods. Classic model-based approaches, such as VAR models (Lütkepohl, 2005), assume linear dynamics and require a predefined maximum lag, which can bias causal assessment. Extensions with sparse penalties (Tank et al., 2021) mitigate this but remain limited by linear assumptions. Non-model-based methods (Lusch et al., 2016; Vicente et al., 2011; Amblard and Michel, 2011) relax linearity, enabling nonlinear causal discovery, but often suffer from high variance and poor scalability in high dimensions (Runge, 2018).

Neural networks have shown strong performance in multivariate time-series forecasting (Yu et al., 2018; Li et al., 2017), yet their application to causal discovery faces two challenges: (1) joint modeling of many variables leads to excessive parameters and computational cost; and (2) the black-box

nature of deep models limits interpretability. To address this, Neural Granger Causality (NGC) (Tank et al., 2021) introduces component-level networks to reduce parameters and improve interpretability. However, RNNs in NGC capture only overall causal strength and impose restrictive assumptions on lag effects, limiting robustness across varying temporal dependencies.

To address these challenges, we propose the Dynamic Regularized Granger Causality Learning model (DRMLP). This model establishes a channel between the recurrent and linear network through sampling causal graph to assist in lag selection. Furthermore, we design a dynamic input weight hierarchical penalty strategy to enhance the accuracy of causality discovery across different lags in the linear network. More specifically, impose hierarchical, lag-aware sparsity on the neural network first layer via proximal updates, encouraging 'near lag first, far lag if necessary'. The main contributions of this paper are as follows:

1. A dual-branch neural network architecture: the first branch is a linear causal discovery network for univariate modeling. It extracts causal relations and applies Gumbel-Softmax sampling to generate processed time series data, which are then used to optimize the training of recurrent networks. The second branch is an MLP network equipped with a dynamic sparsity regularization strategy.

2. A dynamic sparse penalty strategy based on hierarchical group Lasso regularization: enabling the model to learn the numerical relationships between different lags and coordinate the strengths of causal relationships across various lagged variables.

3. For empirical evaluation, we validate the effectiveness of the proposed model on representative simulated datasets and realistic data systems, achieving excellent results compared to several advanced Granger causality discovery methods.

## 2 BACKGROUND AND RELATED WORK

The Granger causality method(Granger, 1969) tests the ability of one time series to predict another, making it widely applicable in the analysis of causal relationships in time series data. Initially, Granger causality was assumed to operate within linear models, with causal structures identified through fitting Vector Autoregression (VAR) models. This concept has since been extended to accommodate nonlinear scenarios(Marinazzo et al., 2008). Furthermore, due to its high compatibility with deep neural networks, Granger causality research were expanded to analyze more complex data, facilitating the examination of deeper or confounded causal relationships within time series.

Granger causality analysis can be divided into model-based and non-model-based methods. Most model-based methods assume linear relationships and utilize autoregressive models(Lozano et al., 2009b). These methods posit that past values of a series have a linear effect on the future values of a target series, where non-zero coefficients quantify the magnitude of the Granger causal effect. Techniques such as Lasso(Tibshirani, 1996) or group Lasso, which induce sparse regularization, help extend linear Granger causality in autoregressive models to high-dimensional scenarios(Lozano et al., 2009b;a). However, the assumption of linearity may lead to misunderstandings of actual nonlinear relationships and could produce inconsistent estimates of underlying structures due to oversimplification.

Non-model-based methods overcome the limitations of linear assumptions by addressing nonlinear dependencies between observed variables, making minimal assumptions about potential relationships. Examples include transfer entropy(Vicente et al., 2011) and directed information(Amblard and Michel, 2011). Results from these methods may have high uncertainty due to degrees of freedom and require large amounts of data, making them less suitable for situations with significantly increased dimensionality(Runge et al., 2012).

Recent advances in deep learning have inspired neural approaches to causal discovery. DYNOTEARS (Pamfil et al., 2020) extends score-based methods to SVAR models but remains restricted to linear VAR structures. Neural regularization techniques (Wu et al., 2020; Xu et al., 2019) and pairwise Granger tests (Singh et al., 2022) aim to capture nonlinear dependencies. Further, non-linear Granger extensions with MLPs and RNNs (Tank et al., 2021) and unified neural frameworks such as NTiCD (Absar et al., 2023) explore richer dynamics. The CUTS family (Cheng et al., 2023; 2024) introduces alternating discovery–imputation phases and later incorporates graph neural net-

works for irregular time series. Despite these advances, existing methods often suffer from either linear assumptions, limited lag modeling, or scalability challenges.

In terms of interpretability, Neural Granger Causality(Tank et al., 2021) employs MLPs and RNNs to derive interpretable nonlinear Granger causality by analyzing the parameters of neural networks. The key aspect of obtaining interpretable causality lies in the independent models of univariate output sequences. Horvath et al.(Sultan et al., 2022) introduced a learning kernel function called LeKVAR and a mechanism that decoupling time lags and individual time series, achieving delayed lag selection and causal interpretation, thus providing improved scalability.

## 3 GRANGER CAUSALITY MODEL

Assume that the variable $x_t \in \mathbb{R}^p$ is a p-dimensional stationary time series, and that the data is observed over a specific time period $1 \leq t \leq T, t \in \mathbb{Z}$ . The linear Granger causality of time series is generally studied using the Vector Autoregressive Model (VAR)(Lütkepohl, 2005). The true data $x_t$, is considered to be a linear combination of the past K lagged values, as shown in Equation equation 1.

$$x_t = \sum_{k=1}^{K} A_k x_{t-k} + e_t \tag{1}$$

where $K$ denotes maximum order of time lag. That is, when $k > K$ , $x_{t-k}$ will not affect $x_t$ in any dimension. $A_k \in \mathbb{R}^{p \times p}$ is a square matrix used to indicate how the values of each dimension of the sequence affect the value at the current moment when the lag is $k$. $e_t \in \mathbb{R}^p$ is the noise error. The potential influence of the value at each lag order $x_{t-k}$ on $x_t$ is obtained through the $A_k$ transformation of the value. Then, after performing accumulation and adding noise operations, the value at the current moment is obtained.

In fields with high-dimensional time series, the data often do not satisfy the linear relationships. The definition of non-linear Granger causality is defined in Definition 3.

**Definition 3.** In the multi-dimensional time series $x$, if for all $(x_{<t1}, \ldots, x_{<tp})$ and $x\prime_{<tj} \neq x_{<tj}$ , it satisfies

$$g_i\left(x_{<t1}, \ldots, x_{<tj}, \ldots, x_{<tp}\right) = g_i\left(x_{<t1}, \ldots, x\prime_{<tj}, \ldots, x_{<tp}\right) \tag{2}$$

It is said that there is no Granger causality between variable $j$ and variable $i$, that is, $g_i$ is invariant with respect to $x_{<tj}$ . If a certain pair of $i$ and $j$ does not satisfy the conditions, it is said that there is Granger causality between variable $j$ and variable $i$.

The problem we investigated is to learn a global causal graph $G$ corresponding to the time series when the input time series is $x_{<tj}$ , using a combined model of linear and recurrent networks. Where applicable, the global causal graph can be extended to a lagged causal graph $G'$. The trained model can subsequently be utilized for time series prediction tasks and the construction of sparse models.

## 4 DRMLP

The proposed model consists of two main parts: the linear causal discovery network and the sampling recurrent network.

The linear causal discovery network fits, for each variable, the causal strength of all other variables' $K$-order lagged values in the time series to its current value. The sampling recurrent network independently samples a causal graph for each variable and generates predictions for the time series.

The training of the model is carried out alternately. For the linear causal discovery network, the input weights are extracted according to the lag order to form a regularization term. In this process, we propose a dynamic regularization strategy that enforces sparsity on the weights. The single-variable causal graph of the target variable is derived from the input weights of the linear network and subsequently applied to the original sequence data. The recurrent network then iteratively outputs the prediction error terms. The training objective combines a structured regularization term with

a prediction fitting term, guiding the model to gradually converge toward accurately capturing the underlying causal mechanisms, as shown in Figure1.

## 4.1 Linear Causal Discovery Network

In the linear component of our method, a separate network model for each variable is established, as

$$x_{ti} = g_i\left(x_{<t1}, \ldots, x_{<tp}\right) + e_{ti} \tag{3}$$

where $x_{ti}$ represents the value of time series $x_i$ at time $t$, and $e_{ti}$ denotes the corresponding random noise at that time. The function $g_i$ specifies how the past $t$ values of the multidimensional time series $x$ are mapped to the individual series $i$. For each variable in the multidimensional series, the same MLP is created, and then all networks are combined in parallel to produce the output. We employ a one-dimensional convolutional network instead of a fully connected linear structure. In the first layer of MLP, i.e., the input layer, we utilize $H$ convolutional kernels of size $p \times K$, where $H$ is the configurable number of hidden units, $p$ is the dimension of the input multidimensional time series, and $K$ is the predetermined maximum possible time lag.

This structure simulates the function $g_i(\cdot)$ in equation 3. The network parameters are determined by the weights $\mathbf{W}$ and biases $\mathbf{b}$ of each layer, where $\mathbf{W} = \{W^1, \ldots, W^L\}$, $\mathbf{b} = \{b^1, \ldots, b^L\}$. $L$ represents the total number of layers. The interpretability of the MLP primarily resides in the input weights. The weight structure of the input layer is represented as a 3-dimensional tensor of size $H \times p \times K$, denoted as $W^1 \in \mathbb{R}^{H \times p \times K}$. We decompose the input weights into $W^1 = \{W^{11}, \ldots, W^{1K}\}$, where $W^{1k} \in \mathbb{R}^{p \times H}$, $k = 1, \ldots, K$. The rest network weights can be represented as $W^l \in \mathbb{R}^{H \times H}(l = 2, \ldots, L-1)$, $W^L \in \mathbb{R}^H$, $b^l \in \mathbb{R}^H(l = 1, \ldots, L-1)$ and $b^L \in \mathbb{R}$. The hidden vector resulting from the input data at time t can be expressed as

$$h_t^1 = \sigma\left(\sum_{k=1}^{K} W^{1k} x_{t-k} + b^1\right) \tag{4}$$

where $\sigma$ is the activation function, such as typical logistic function or ReLU. The hidden vectors in subsequent layers are denoted as $h_t^l$, and their calculations also utilize the same activation function $\sigma$, as shown in Equation equation 5.

$$h_t^l = \sigma\left(W^l h_t^{l-1} + b^l\right) \tag{5}$$

After passing through $L - 1$ hidden layers, the output univariate sequence $x_{ti}$ is represented as a linear combination of all units in the final hidden layer, as

$$x_{ti} = W^L h_t^{L-1} + b^L + e_{ti} \tag{6}$$

where the error term $e_{ti}$ is modeled using a zero-mean Gaussian noise distribution.

## 4.2 Sampling-based Causal Discovery Network

We apply Bernoulli sampling to the univariate causal graphs extracted from the input weights of MLP. The sampling process enables us to overlay the causal graphs onto series data, ensuring that the input data for the LSTM network retains only the information that has a causal impact on the target variable, while minimizing the influence of other variables. Once the data has been processed, it is fed into the corresponding LSTM network for computation.

Through the supervisory role of LSTM, the model learns the causal relationships between dimensions of series data from both static sample data and time-dependent data perspectives.

### 4.2.1 Bernoulli Sampling of Unidimensional Causal Graph

In the MLP network, each univariate network extracts a causal graph denoted as $G_i \in \mathbb{R}^p$, where $i = 1, \ldots, p$. A complete causal graph $G$ can be obtained by concatenating multiple univariate causal graphs. The elements $c_{ij}$ in the causal graph $G_i$ indicate the degree of causal influence that variable $j$ has on variable $i$. We optimize $G_i$ using Gumbel-softmax sampling(Zhang and Ghanem, 2018), as

$$s_{ij} = \frac{e^{(\log(c_{ij})+g)/\tau}}{e^{(\log(c_{ij})+g)/\tau} + e^{(\log(1-c_{ij})+g)/\tau}} \tag{7}$$

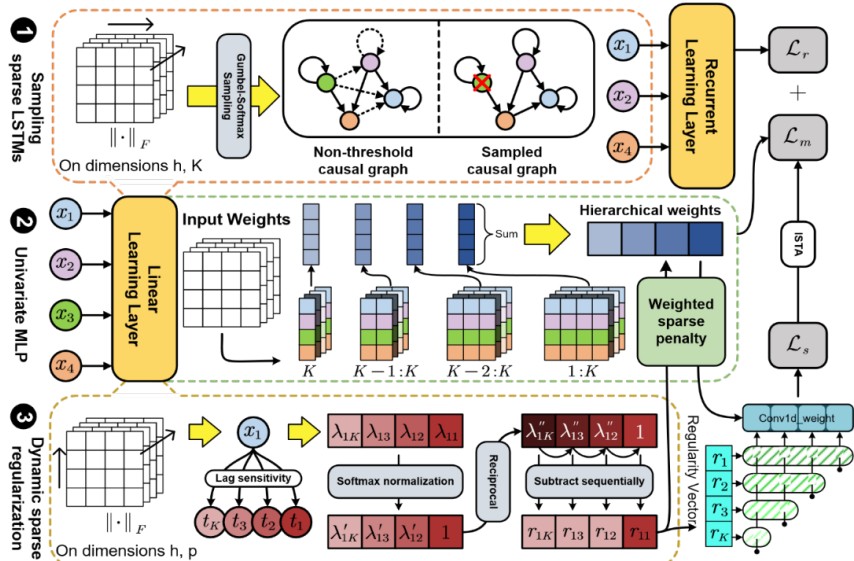

Figure 1: overall model structure

where $g = -\log(-\log(u))$ and $u \sim \text{Uniform}(0,1)$. The variable $\tau$ represents a temperature coefficient. A smaller $\tau$ value results in sampling that is closer to Bernoulli sampling.

By selecting such a soft sampling approach, after performing the softmax expectation operation, dimensions with samples close to 1 will retain original series data, preserving their influence on the target variable. Conversely, dimensions with samples close to 0 will be significantly reduced, thereby greatly diminishing their impact on the target variable within the LSTM.

Using Gumbel-softmax sampling method allows the sampling operation to backpropagate through the loss computation, ensuring that the loss information from the LSTM network can be transmitted back to the input weights of MLP.

### 4.2.2 CAUSAL GRAPH COVERAGE DATA

After obtaining the univariate causal graph through sampling, we perform a mask operation on the original series. It involves taking the causal graph vector and performing a dot product with the original data vector at each time step, resulting in a new set of series data known as the sampled data, which is the input of LSTM.

During the training process, the closer the sampling results of the causal graph are to the true causal graph, the more the sampled data will approximate the true set of causal variables for the target variable, leading the input data to yield predictions that are closest to the original values.

Additionally, the sampling operation introduces a form of "random perturbation" into the training process, which helps prevent the network from getting trapped in local solutions formed by spurious causal relationships. It enhances the robustness of the model, allowing it to generalize better to unseen data.

### 4.2.3 LOSS FUNCTION AND DYNAMIC SPARSE REGULARIZATION

The loss function of the model consists of three components: the prediction error from MLP and LSTM network and a dynamic sparse regularization term for the input weights:

$$\mathcal{L}(\mathbf{W}) = \sum_{t=K}^{T} \left( x_{it} - g_i \left( x_{(t-K):(t-1)} \right) \right)^2 + \sum_{t=K}^{T} \left( x_{it} - l_i \left( x_{(t-K):(t-1)} \right) \right) + \lambda \rho \left( W^1 \right) \quad (8)$$

where $g_i(\cdot)$ represents the function of MLP for variable $i$, while $l_i(\cdot)$ denotes the function of LSTM for the same variable. The term $\lambda \rho(\cdot)$ refers to the regularization with coefficients applied to the input weights of MLP.

We adopt a hierarchical group Lasso penalty strategy that updates the regularization coefficients at fixed frequencies. We set distinct regularization coefficients for different time lags based on the overall dependency degree of all other variables at specific lags obtained during the training process, and we add a constant to control the overall regularization effect.

By calculating the F-norm of the input weights along the 0-th and 1-st dimensions and performing normalization, we obtain the dimension-averaged lag dependency vector $\lambda_i = (\lambda_{i1}, \lambda_{i2}, \ldots, \lambda_{iK})$, where $i = 1, \ldots, p$ and $k = 1, 2, \ldots, K$. $\lambda_{ik}$ represents the average dependency degree of variable $i$ on the values from the past $k$ time points, with the averaging operation executed across all variable dimensions. The hierarchical adjustment of $\lambda_i$ results in the dimension-averaged hierarchical lag penalty vector. The calculation process is described as

$$\boldsymbol{\lambda}'_i = \frac{1}{\boldsymbol{\lambda}_i}, \quad \boldsymbol{\lambda}''_i = \frac{\boldsymbol{\lambda}'_i}{\lambda'_{i1}}$$

$$r_{i1} = \lambda''_{i1}, \quad r_{ik} = \lambda''_{ik} - \lambda''_{i(k-1)}, \quad k = 2, \cdots, K$$

$$\boldsymbol{r}_i = (r_{i1}, \cdots, r_{iK})$$

where $\lambda_i$ is the dimension-averaged lag dependency vector, while $\lambda'_i$ and $\lambda''_i$ are intermediate variables. $\mathbf{r}_i$ represents the dimension-averaged hierarchical lag penalty vector for variable $i$. The regularization $\rho(W^1)$ with the dynamic hierarchical penalty is described as

$$\rho\left(W^1\right) = \sum_{j=1}^{p} \sum_{k=1}^{K} r_{ik} \left\| W_j^{1k}, \cdots, W_j^{1K} \right\|_F \tag{9}$$

where $W_j^{1k}$ represents the column of input weights corresponding to input variable $j$ at lag $k$, $r_{ik}$ is the $k$-th term of $\mathbf{r}_i$, and $\|\cdot\|_F$ denotes the operation of calculating the F-norm, as illustrated in the third part of Figure 1.

### 4.2.4 OPTIMIZATION OF LOSS FUNCTION WITH REGULARIZATION

We employ the iterative soft-thresholding shrinkage algorithm (ISTA) (Zhang and Ghanem, 2018)to optimize the loss function. ISTA is a specific form of the proximal gradient descent algorithm, where the loss function consists of the mean squared error of the predictions plus a sparse regularization term with coefficients. This formulation encourages certain rows or columns of the weight matrix to fall within the threshold range, effectively achieving precise zero values.

When using ISTA for target optimization, we implement a line search approach to ensure that the loss function converges to a local minimum. The weights are randomly initialized from a standard normal distribution as $\mathbf{W}^{(0)}$, and the optimization algorithm iteratively updates the weights starting from $\mathbf{W}^{(0)}$ as

$$\mathbf{W}^{(n+1)} = \text{prox}_{d^{(m)}\lambda\rho}\left(\mathbf{W}^{(n)} - d^{(n)}\nabla\mathcal{L}\left(\mathbf{W}^{(n)}\right)\right) \tag{10}$$

where $\mathbf{W}^{(n)}$ represents the weights at the $n$-th iteration step. The step size $d^{(n)}$ for the $n$-th iteration. $\mathcal{L}(\mathbf{W})$ denotes the prediction error. The operator $\text{prox}_{\lambda\rho}(\cdot)$ is the proximal operator concerning $\rho(\cdot)$ and $\lambda$.

The dynamic sparse regularization are only applied to the input weights of MLP, while the iteration step sizes for other layers of MLP and all layers of LSTM are fixed values. The approximate step size is obtained through a hierarchical weighted soft-thresholding operation on the input weights, as

$$\text{prox}_{d^{(m)}\lambda\rho}\left(W_{:k}^1\right) = \left(1 - \frac{r_k\lambda d^{(m)}}{\|W_{:k}^1\|_F}\right)_+ W_{:k}^1 \tag{11}$$

where $W_{j \cdot k}^{1 \cdot}$ represents the portion of the input weights corresponding to the previous $k$ lags, and $r_k$ is the average hierarchical lag penalty corresponding to each lag order. $(\theta)_+ = \max(0, \theta)$. Since F-norm of the weights yields non-negative values, we only need to consider the case where the weights fall within the threshold range from positive values.

The approximate step size for the input weights is calculated by iteratively applying the group soft-thresholding operation to the penalty function across different lag ranges.

## 5 EXPERIMENT

We validated the proposed model DRMLP on both simulated datasets and datasets inspired by real-world scenarios. The results of the causal graph learning were compared with the experimental results of several mainstream baseline methods to verify the effectiveness of method proposed. Additionally, we set different regularization parameters for the model to examine their impact on the training performance.

### 5.1 DATASET

**VAR data.** We generate synthetic multivariate time series from linear vector autoregressive models $VAR(d)$ with dimensions $p = 10$ and lengths $T \in \{200, 500, 1000\}$ and consider $d = 2, 3$ to construct $VAR(2)$ and $VAR(3)$ datasets for evaluation. (Lütkepohl, 2005)
**Lorenz-96 data.** We further evaluate our model on the Lorenz-96 benchmark(Lorenz, 1996)., a nonlinear chaotic dynamical system widely used for time series analysis. We generate multivariate series with dimension $p = 10$, forcing constants $F \in \{10, 20\}$, and sampling rate $\Delta t = 0.05$, resulting in nonlinear time series with complex temporal dependencies.
**DREAM-3.** We also validate the proposed model on the real-world inspired time gene expression dataset DREAM-3(Prill et al., 2010).

### 5.2 BASELINE

We select the following four mainstream baseline methods for comparison:

1. **Neural Granger Causality Method (NGC)** (Tank et al., 2021): This method utilizes utilizes a combination of multilayer perceptrons (MLP) and recurrent neural networks (RNN) with group penalties to infer Granger causality. In our experiments, we employ the MLP model for the VAR dataset and the RNN model for the Lorenz-96 dataset.

2. **PCMCI** (Runge et al., 2019): This method employs conditional independence testing to detect nonlinear Granger causality. It is designed to identify causal relationships in complex systems.

3. **Economy-SRU** (Khanna and Tan, 2019): This method uses a component time series forecasting model based on Statistical Regression Units (SRU) for nonlinear modeling. By designing a small number of trainable parameters, it enhances the model's robustness against overfitting when predicting data.

4. **CUTS** (Cheng et al., 2023): This method learns causal relationships in data through an alternating process of causal discovery and latent data prediction. It is capable of simultaneously discovering causality and filling in missing values in irregular datasets.

### 5.3 METRICS

In terms of quantitative evaluation, to verify the accuracy of the learned causal graph in reconstructing the ground truth, we used the Area Under the Receiver Operating Characteristic Curve (AUROC) as the evaluation metric.

### 5.4 EXPERIMENTAL RESULTS AND ANALYSIS

#### 5.4.1 AUROC IN CAUSAL GRAPH LEARNING

As shown in Table1,The method named "DRMLP-s" in the table represents an ablation model that removes the recurrent network module based on DRMLP. The values presented in Table1 (e.g., 99.86±0.12) are AUROC values, where the complete value corresponds to "0.9986±0.0012" in decimal form or 98.66% ± 0.12% in percentage form.

The experimental results indicate that the DRMLP method generally achieved the best results on different lengths of VAR data and Lorenz-96 data. Compared to NGC, DRMLP shows an improvement of approximately 1% to 5% in metrics on longer sequences, which can be attributed to the effect of

Table 1: Comparison of Models on Different Datasets (Accuracy %)

| | VAR(2) | | | VAR(3) | | |
|---|---|---|---|---|---|---|
| Model | T=200 | T=500 | T=1000 | T=200 | T=500 | T=1000 |
| NGC | 96.01±2.14 | 91.98±3.87 | 98.52±1.31 | 90.91±4.15 | 97.53±2.82 | 98.74±0.98 |
| PCMCI | 71.40±4.93 | 72.30±5.73 | 72.19±3.25 | 65.89±5.23 | 71.30±4.85 | 71.92±3.61 |
| eSRU | 87.74±6.71 | 89.97±5.72 | 91.06±4.86 | 81.25±7.05 | 86.55±5.48 | 90.38±2.29 |
| CUTS | 98.92±0.84 | 99.05±1.03 | 100.00±0.00 | 98.05±1.64 | 99.91±0.03 | 99.99±0.02 |
| DRMLP-s | 99.17±0.98 | 100.00±0.00 | 100.00±0.00 | 97.56±2.05 | 100.00±0.00 | 100.00±0.00 |
| **DRMLP** | **99.86±0.12** | **100.00±0.00** | **100.00±0.00** | **98.86±1.42** | **100.00±0.00** | **100.00±0.00** |

| | Lorenz-96 (F=10) | | | Lorenz-96 (F=20) | | |
|---|---|---|---|---|---|---|
| Model | T=200 | T=500 | T=1000 | T=200 | T=500 | T=1000 |
| NGC | 93.88±1.64 | 98.56±0.51 | 99.15±0.33 | 84.35±3.31 | 92.22±3.72 | 92.82±2.60 |
| PCMCI | 85.95±4.55 | 91.65±3.54 | 95.72±2.23 | 80.87±5.21 | 86.56±3.18 | 86.39±2.71 |
| eSRU | 87.42±3.98 | 96.45±3.06 | 97.51±1.16 | 90.30±3.63 | 97.57±1.43 | 98.43±1.74 |
| CUTS | **95.12±1.13** | 99.64±0.51 | 100.00±0.00 | 89.20±0.73 | 93.36±0.82 | 97.15±2.94 |
| DRMLP-s | 93.63±0.55 | 98.64±0.86 | 100.00±0.00 | 90.51±1.48 | **95.66±1.62** | 98.01±1.57 |
| **DRMLP** | 94.91±0.96 | **100.00±0.00** | **100.00±0.00** | **92.00±0.97** | 95.49±2.30 | **98.80±0.45** |

dynamic regularization. The correct selection of lag values allows the model to accurately quantify the dependence of the target variable on the causal variables, thereby precisely excluding the minor influences of irrelevant variables. In fact, the incorrect causal relationships chosen by NGC often have only a small weight compared to the correct ones across any lag values, and traditional regularization methods are not effective in eliminating the influence.

### 5.4.2 AUROC with Different Dependency Coefficients

The dependency coefficient $p$ of the VAR data represents the number of variables driving each variable. A $p$ value of 0.2 indicates that the number of causal variables accounts for 20% of the total number of variables. As the $p$ value increases, the number of interdependent variables increases, which makes it more complex and challenging for the model to accurately discover the correct causal graph. We generated three types of VAR(3) data with dependency coefficients of 0.2, 0.3, and 0.4, and the learning results are presented in Table 2

Table 2: Accuracy (%) under different dependency strengths $p$

| | p = 0.2 | | | p = 0.3 | | | p = 0.4 | | |
|---|---|---|---|---|---|---|---|---|---|
| Model | T=200 | T=500 | T=1000 | T=200 | T=500 | T=1000 | T=200 | T=500 | T=1000 |
| NGC | 90.91±4.15 | 97.53±2.82 | 98.74±0.98 | 81.18±3.16 | 90.38±3.91 | 93.84±3.27 | 75.85±5.02 | 82.84±4.55 | 87.82±3.67 |
| PCMCI | 65.89±5.23 | 71.30±4.85 | 71.92±3.61 | 57.52±4.83 | 54.95±6.50 | 54.91±5.81 | 56.49±7.42 | 51.95±7.75 | 53.49±7.78 |
| eSRU | 81.23±7.16 | 86.22±5.49 | 90.10±2.66 | 70.75±7.33 | 75.68±6.73 | 82.26±4.11 | 63.95±7.48 | 68.60±5.13 | 73.99±8.05 |
| CUTS | 98.97±1.64 | 99.91±0.03 | 99.99±0.02 | 82.58±4.79 | 95.33±0.99 | 94.45±2.28 | 75.91±6.73 | 83.43±1.65 | 92.59±3.31 |
| DRMLP-s | 97.56±2.05 | 100.0±0.00 | 100.0±0.00 | 85.64±4.09 | **99.43±0.75** | **100.0±0.00** | 76.06±5.96 | **91.99±1.38** | **95.72±0.51** |
| **DRMLP** | **98.86±1.42** | **100.0±0.00** | **100.0±0.00** | **86.57±6.07** | 90.63±5.08 | 94.14±5.34 | **78.67±7.13** | 85.82±2.27 | 88.21±6.77 |

The experimental results indicate that as the dependency coefficient increases, the accuracy decreases for all methods. When each variable has only two causal variables, DRMLP can accurately predict the correct causal graph. For more complex causal graphs, DRMLP still outperforms other methods. With sequence lengths of 500 and 1000, the ablation model, benefiting from a sufficient number of training samples, achieved the best results. Meanwhile, the complete model with LSTM was able to better capture temporal dependencies even with shorter sequence lengths, showing superior performance.

### 5.4.3 LAG SELECTION ON VAR(3)

In the simulation of $VAR(3)$ data, each variable depends on its past three lags, with equal causal coefficients of 0.2. We generate sequences of length $T = 1000$ for causal graph learning. To evaluate whether the model can exclude higher-order lags, we set the convolution kernel size to 5. The comparison baseline is cMLP with hierarchical Lasso, and we illustrate results on the 2-nd and 5-th variables among the 10 dimensions (Figure 3).

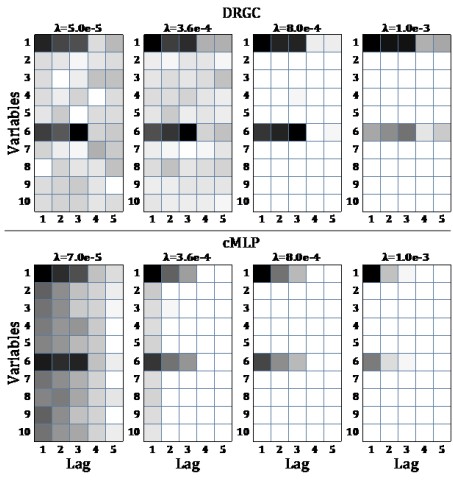
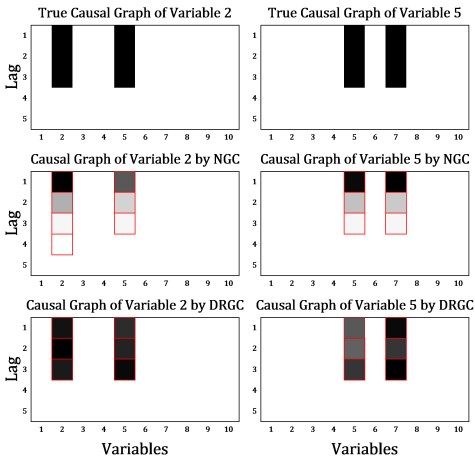

Figure 2: Lagged causal graph on VAR(3) of DRMLP and cMLP with different dynamic regularization coefficients

Figure 3: Lagged causal graph on VAR(3) of DRMLP and cMLP

NGC suffers from the rigid hierarchical penalty, which weakens true causal effects and fails to suppress irrelevant higher-order lags. In contrast, DRMLP recovers consistent causal strengths across the first three lags, matching the true $VAR(3)$ mechanism.

### 5.4.4 LAG SELECTION ON DISTINCT REGULARIZATION COEFFICIENTS

We further test $VAR(3)$ with different dynamic regularization coefficients $\lambda$ and compare the learned lagged causal graphs with the ground truth (Figure 2). When $\lambda$ is appropriate, both NGC and DRMLP recover the correct structure, while DRMLP assigns lag strengths more reasonably. NGC tends to overestimate lag orders for small $\lambda$ and suppress later lags for large $\lambda$, whereas DRMLP remains stable and preserves relative magnitudes. Excessively large $\lambda$ drives all weights to zero, highlighting the need for cross-validation to choose $\lambda$.

## 6 CONCLUSION

Granger causality learning method that integrates dynamic hierarchical sparse penalties with linear and recurrent networks. To enhance interpretability, we used separate linear and recurrent networks for each variable. The recurrent network extracts temporal dependencies, supervising and refining causal relationships identified by the linear network. Sparse penalties, based on average dependency levels at various lags, improve the model's accuracy in selecting causal relationships. Experimental results on simulated and gene regulatory network datasets show superior performance and stability, even under varying sparse penalty parameters. Future work will explore parameter selection without true causal information and investigate real-world datasets to enhance model generalizability and causal verification.

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

# A  APPENDIX

## A.1  CASE STUDY: DREAM-3

This dataset presents a challenging nonlinear dataset for Granger causality testing. It simulates con-
tinuous gene expression and regulatory dynamics, with multiple hidden factors that are unobserved.
DREAM-3 contains five simulated datasets with different true causal relationships. Each dataset
consists of four segments of data with 10 nodes, each segment having 21 sampling time points.
Different sequences are concatenated to form a total sequence of length 84 for experimentation.

The DREAM-3 dataset is a challenging nonlinear dataset used for rigorously comparing Granger
causality detection methods. It contains three Yeast (Y.) datasets and two E. Coli (E.C.) datasets. In
terms of variable count and underlying structural complexity, these datasets represent a limited data
system, thus posing a significant test for the capabilities of causality learning methods. We applied
DRMLP to learn from these five datasets while using cMLP and cLSTM as comparison models.
Given the short sequence lengths of the DREAM-3 data, we set the maximum lag $L$ to 2 and the
number of hidden layer units to 10 to reduce the model's size and accelerate the training process.
The same settings were applied to the other methods. The evaluation results across the five time
series datasets are presented in Table 3, with the corresponding ROC curves shown in Figure4.

Table 3: AUROC for DRMLP, cMLP, and cLSTM models on DREAM-3

| Model | Yeast 1 | Yeast 2 | Yeast 3 | Ecoli 1 | Ecoli 2 |
|-------|---------|---------|---------|---------|---------|
| cMLP  | 0.5722  | 0.5629  | 0.5553  | 0.5741  | 0.5855  |
| cLSTM | 0.5933  | 0.5800  | 0.5525  | 0.5132  | 0.5647  |
| DRMLP | **0.6568** | **0.6485** | **0.6067** | **0.6236** | **0.6070** |

From Table 3, it is evident that DRMLP outperformed both cMLP and cLSTM across all five
datasets. Notably, DREAM-3 with 10 nodes only contains four time series of length 21, result-
ing in a limited number of training samples, which makes it challenging for the models to learn the

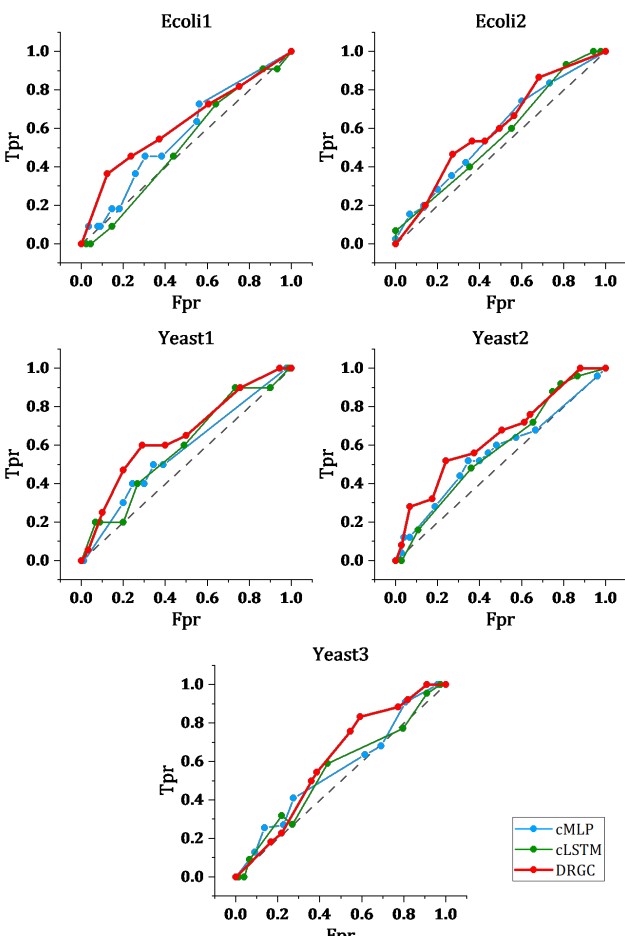

Figure 4: ROC for DRMLP, cMLP, and cLSTM models on DREAM-3

correct causal graph. Consequently, the AUROC values for all models were relatively low. However, DRMLP achieved a level of 60%. Our method effectively combines the advantages of linear and recurrent networks, allowing for the efficient reduction of the influence of irrelevant variables during the execution of sparse penalties, thereby concentrating the weights on the corresponding true causal variables.

## A.2 LLM USAGE

We used large language models (LLMs), specifically OpenAI's ChatGPT, as a supporting tool in the preparation of this manuscript. The LLM was employed **only for language-related assistance**, including:

- rephrasing and condensing sentences to improve readability,
- translating text between Chinese and English, and
- minor grammar and style corrections.

The LLM was **not used** for research ideation, theoretical development, experimental design, analysis, or the creation of novel scientific content. All research contributions, methodology, results, and interpretations presented in this paper are solely the work of the authors. The authors take full responsibility for the content of this paper.

