# OpenReview forum: "DRMLP: Dynamic Regularized Multi-Layer Perceptron for Neural Granger Causality Discovery with Adaptive Temporal Penalties"
_ICLR.cc/2026/Conference — ICLR 2026 Conference Withdrawn Submission_

### Official Review · Reviewer_ateh · 2025-10-30

**Soundness:** 3
**Presentation:** 2
**Contribution:** 3
**Rating:** 6
**Confidence:** 2

**Summary:**

This paper presents DRMLP, a Dynamic Regularized Multi-Layer Perceptron framework for discovering Granger causal structure in multivariate time series. DRMLP introduces a dual-branch neural architecture, combining a linear (MLP-based) causal discovery path with a recurrent (LSTM-based) sampling strategy, and applies an adaptive, hierarchical sparse penalty on input convolutional weights to improve temporal lag selection. The paper claims improved robustness to long-range dependencies and enhanced accuracy in causal discovery, demonstrated on both simulated (VAR, Lorenz-96) and real-world-inspired (DREAM-3) datasets, with empirical comparisons to state-of-the-art baselines.

**Strengths:**

The dual-branch design, combining an MLP for causal parameter inference and an LSTM for temporal supervision via masked sampling, is a creative architectural decision.

The introduction of a lag-sensitive, dynamically updated group Lasso regularization offers granularity for lag selection that addresses some classic weaknesses of conventional neural Granger models.

Lag selection results are directly visualized and compared in Figures 2 and 3, showing how DRMLP distinguishes correct lags versus baselines, which this supports the claim of better selectivity and makes the method more transparent.

The regularization path, especially the hierarchical penalty formulation and its implementation via the proximal operator and ISTA, is clearly specified with comprehensive notation

**Weaknesses:**

The experiments are mostly restricted to relatively low-dimensional simulated data ($p=10$), with only DREAM-3 providing a real-world inspired, but still highly constrained, testbed.

There is insufficient empirical ablation on kernel size, number of hidden units, or how these impact interpretability and discovery power.

The manuscript does not sufficiently discuss the potential downsides or edge cases where the method may struggle, e.g., data with strong cross-lag nonlinearities not captured by MLP or LSTM, or scenarios where sampling-induced randomness could hurt stability.

Evaluation on DREAM-3 is Somewhat Superficial. The comparison is limited to cMLP and cLSTM; many recent graphical and nonparametric causal methods are absent.

**Questions:**

Given the high sensitivity of lag recovery to the penalty parameter $\lambda$ (as seen in Figure 2), can you provide systematic guidance or a robust selection protocol for $\lambda$?

How does DRMLP scale computationally with the number of variables (e.g., p > 100)?

Does the proposed dynamic penalty require manual tuning, or could it be learned jointly with network parameters?

In Table 3 and Figure 4, all methods produce relatively low AUROC. Why?

---

### Official Review · Reviewer_Zpi5 · 2025-10-31

**Soundness:** 2
**Presentation:** 2
**Contribution:** 2
**Rating:** 2
**Confidence:** 3

**Summary:**

This paper proposes a two level hierarchy for learning Granger Causal Networks from observational data as a dynamically regulated multi-layer perceptron using: (i) a linear causal discovery network is utilized to extract causal relations from sampled weight data; (ii) hierarchical regularization strategy is introduced to optimize the weights of the network.  They have used synthetic datasets and some real world datasets to showcase how their approach can learn rich granger causal networks in different contexts.

**Strengths:**

The paper solves an important problem and I like the fact that the authors aim to decouple the problem into two separate steps and maintain the conventional simplicity of linear causal pathways and combine that with careful sampling and adjustment of weights.

**Weaknesses:**

1. The underlying techniques may be well known in the literature and the paper comes across as an incremental amalgamation of known ideas.

2. The real-world datasets used to evaluate the algorithms seem too old.

3. The baseline comparisons do not come across as state of the art.

4. The paper could benefit from an array of theoretical contributions which can outline under what constraints is their proposed approach going to yield high quality granger causal networks which are more intuitive, explainable and have high confidence in the discovered network links.

**Questions:**

I appreciate the simplicity of the approach and the decomposition of the problem into two steps and how this decomposition is able to provide explainable granger causal network models that are able to perform better than previous neural granger causal models.

The paper is ignoring several basic works from the ML literature on Graph attention networks, Graph neural networks, Conditional Granger Causal Networks etc. Many of the references in the work are beyond the conventional ML literature which is great but the comparisons should also match with some of the core approaches covered in the ML space. Here are some refs I found via Google Scholar (may not be the latest pointed works):

Jiaxuan You, Rex Ying, Xiang Ren, William L. Hamilton, and Jure Leskovec. Graphrnn: A deep
generative model for graphs. CoRR, abs/1802.08773, 2018. URL http://arxiv.org/abs/1802.08773.

Petar Veliˇ ckovi´ c, Guillem Cucurull, Arantxa Casanova, Adriana Romero, Pietro Liò, and Yoshua
Bengio. Graph attention networks. In International Conference on Learning Representations,
2018.

Learning Conditional Granger Causal Temporal Networks
Ananth Balashankar, Srikanth Jagabathula, Lakshmi Subramanian Proceedings of the Second Conference on Causal Learning and Reasoning, PMLR 213:692-706, 2023.

Please do a more thorough literature review and compare against better baselines in the literature.

The second comment is the baselines need to be carefully chosen and evaluated.

The CUTS paper referred is an arXiv version from 2023 which seems to be unpublished.  Why is the CUTS paper a good framework to compare against for your work?

The other baselines are 2019 or before. I appreciate some of the literature description describing these baselines but it also appears that the paper may have missed out on important references.

The datasets used in the analysis are very old. The VAR dataset is from 2005 and the Lorenz96 dataset is from 1996 and the DREAM3 is a beaten to death dataset for this causal benchmark.

In essence, this subfield has a broad array of papers and the onus is on the authors to also run their experiments on the most appropriate datasets. Using old datasets does not convey a confidence in the methods used.

Finally, the paper may benefit from additional theoretical contributions to strengthen the results.

Under what assumptions and the quality of training data, can the approach discover high quality granger causal network edges?

---

### Official Review · Reviewer_eW8g · 2025-10-31

**Soundness:** 2
**Presentation:** 3
**Contribution:** 2
**Rating:** 2
**Confidence:** 4

**Summary:**

The paper proposes a dual-branch framework for nonlinear Granger causality (GC) discovery in multivariate time series. One branch is a per-variable MLP with hierarchical, lag-aware sparsity applied to the input layer, while the other branch is an LSTM trained on inputs masked by a Gumbel–Softmax–sampled causal graph inferred from the MLP. The two branches are trained alternately. The core assumption is that as the sampled causal graph becomes closer to the true underlying graph, the selected inputs will better approximate the true causal variables of each target, thereby improving the LSTM’s predictive performance.

**Strengths:**

The method is easy to follow, and the overall idea of combining an MLP (primarily for causal discovery) with an LSTM (where a good causal graph also leads to better prediction) is interesting and reasonable. The experiments on synthetic VAR data cover multiple sequence lengths and dependency densities.

**Weaknesses:**

1. I would say the novelty is rather limited. The core components, including per-target networks, group penalties for lagged inputs, and RNN/MLP variants, have already been extensively explored in Neural GC and related prior works cited by the authors.

2. I am particularly concerned about the experimental evaluation, which is quite weak. On one hand, many state-of-the-art (SOTA) neural GC approaches (e.g., [1,2]) have been developed in the past two years, yet the authors only compare their method with approaches from around 2022. In addition, several recent methods are not even discussed in the related work section, which makes the paper appear outdated and lacking in comprehensive context.

[1] Zhou, Wanqi, et al. "Jacobian Regularizer-based Neural Granger Causality." International Conference on Machine Learning, 2024.

[2] Liu, Meiliang, et al. "Kolmogorov-Arnold Networks for Time Series Granger Causality Inference." arXiv preprint arXiv:2501.08958 (2025).

**Questions:**

1. Please include comparisons with state-of-the-art approaches, such as [1,2], and also consider more recent methods published in 2025.

2. Can your method be extended to infer a full time-varying or time–instant-level causal graph (see [3]), rather than relying on a global causal structure?

[3] Assaad, Charles K., Emilie Devijver, and Eric Gaussier. "Survey and evaluation of causal discovery methods for time series." Journal of Artificial Intelligence Research 73 (2022): 767-819.

3. In your abstract, you mention that a limitation of existing approaches is that they typically require modeling each time-series variable separately. However, if I understand correctly, your method also builds a single prediction model for each time-series variable, rather than a unified model (see [1])?

---

### Official Review · Reviewer_4YnP · 2025-11-01

**Soundness:** 2
**Presentation:** 1
**Contribution:** 2
**Rating:** 2
**Confidence:** 3

**Summary:**

### The review

This paper proposes DRMLP, a novel dual-branch neural network for discovering Granger causal relationships in multivariate time series. The model aims to address key limitations of existing neural Granger causality methods, namely the difficulty in modeling long-range or periodic dependencies and the use of static regularization penalties that treat all time lags equally.

The core technical novelty is the dynamic regularized penalty, a hierarchical group Lasso applied to the input weights of the linear MLP. This penalty is updated during training based on the learned dependencies at different lags, allowing the model to encourage near lag first, far lag if necessary. The prediction losses from both branches are combined, allowing the LSTM to supervise the causal structure learned by the MLP.

**Strengths:**

S1. The empirical results reported in the main paper (Table 1) on synthetic VAR and Lorenz-96 datasets are excellent, achieving near-perfect AUROC in many settings and clearly outperforming strong recent baselines like CUTS and NGC. The qualitative lag-selection plots (Figs 2 & 3) also compellingly show the method works as intended on these datasets.

**Weaknesses:**

W1. The paper is extremely difficult to read and understand, bordering on non-reproducible. It critically lacks clarity and reproducability.

 > The architecture design is not well-explained and the interaction between the two branches is confusing. The text describes an "alternate" training process and a complex gradient path (from LSTM loss, through Gumbel-Softmax sampling, back to the MLP weights) that is not clearly detailed. Furthermore, it is not aligned with the Figure 1.

> Dynamic regularization (Sec 4.2.3), is vaguely defined. The "dimension-averaged lag dependency vector" ($\lambda_i$) is the key, but its calculation is described only in text ("calculating the F-norm... along the 0-th and 1-st dimensions") without a precise equation. This makes the core novelty impossible to reproduce. Even worse, there are no justification to make "averaged" lag dependency vectors. Is it safe to just average these vectors?

> The main architecture diagram (Figure 1) is indecipherable. The data flow is unclear, labels are confusing and not aligned with the writing, and the visualization of the dynamic penalty does not clearly map to the text or equations.

W2: Contradictory and Weak Real-World Validation: The paper's empirical strength is severely undermined by the results provided in the appendix.

> The DREAM-3 results (Appendix A.1, Table 3) are very weak, with AUROC values in the ~0.60-0.65 range. This is barely superior to random chance (0.5) and stands in stark contrast to the near-perfect synthetic results. This suggests a significant failure to generalize to more realistic, nonlinear, and short-sequence data. These results are critical and must be included and discussed in the main paper.
> I recommend to check the recent benchmark, CausalTime, which provide more realistic datasets to test your model.

[1] CausalTime: Realistically Generated Time-series for Benchmarking of Causal Discovery, in ICLR'24


W3. The paper is motivated by addressing "long-range dependencies." However, the experiments are conducted on a VAR(3) model with a max lag K=5. This is not a "long-range" dependency. A crucial experiment is missing (e.g., a VAR model with true dependencies at K=20 or K=50) to prove that the dynamic penalty is superior to static penalties in such a scenario. The current experiments are insufficient to validate the central claim.

W4. Incomplete and Potentially Misleading Baselines:

> The paper compares against NGC (Tank et al. 2021) but states, "we employ the MLP model for the VAR dataset." The full NGC model also includes an RNN variant, which is a more direct and powerful competitor for temporal data. The DREAM-3 experiments in the appendix use even weaker baselines (cMLP, cLSTM) and inexplicably omit the stronger baselines from the main paper (CUTS, NGC, ESRU, PCMCI), making the results in Table 3 difficult to contextualize.

**Questions:**

Q1. How do the authors explain the dramatic performance collapse between the synthetic data (Table 1, ~100% AUROC) and the DREAM-3 data (Table 3, ~60-65% AUROC)? This suggests a major generalization problem.

Q2. Why were the main paper's strong baselines (CUTS, NGC, etc.) not included in the DREAM-3 benchmark (Appendix)? Please provide a full comparison on this dataset.

Q3. How does DRMLP perform on a task with true long-range dependencies (e.g., K=50 or K=100)? The current K=5 experiment does not support the claims about modeling long-range dependencies.

Q4. The abstract states existing methods "require modeling each time series variable separately," but Section 4.1 says DRMLP establishes "a separate network model for each variable." This seems to be a direct contradiction. Could you please clarify this point?

---

### Note · Authors · 2025-11-13

I have read and agree with the venue's withdrawal policy on behalf of myself and my co-authors.